# Ultrasound Findings of Fetal Infections: Current Knowledge

Rosita Verteramo, Erica Santi, Francesca Ravennati, Gennaro Scutiero, Pantaleo Greco and Danila Morano *

Department of Morphology, Surgery and Experimental Medicine, Section of Obstetrics and Gynecology, Azienda Ospedaliero-Universitaria S. Anna, University of Ferrara, Cona, 44122 Ferrara, Italy; r.verteramo@ospfe.it (R.V.); erica.santi88@gmail.com (E.S.); francesca.ravennati@gmail.com (F.R.); g.scutiero@hotmail.com (G.S.); pantaleo.greco@unife.it (P.G.)
* Correspondence: moranodanila@gmail.com; Tel.: +39-0532-239517

**Abstract:** Infectious diseases during pregnancy are still a major cause of fetal mortality and morbidity worldwide. The most common teratogenic pathogens are cytomegalovirus (CMV), varicella-zoster virus (VZV), rubeovirus, parvovirus B19, herpes simplex virus (HSV), Toxoplasma gondii, Treponema pallidum and the emergent Zika virus (ZIKV). Ultrasound findings include cerebral anomalies, orbital defects, micrognathia, cardiac defects, hepatosplenomegaly, liver calcifications, abdominal anomalies, skin and limb anomalies, edema, placental and amniotic fluid anomalies and altered Doppler analyses. The classification of ultrasound markers of congenital infections by anatomical region is reported to guide differential diagnosis and prenatal care.

**Keywords:** ultrasound findings; Zika virus; ultrasound fetal anomalies; ultrasound markers; congenital infections

## 1. Introduction

The maternal immune system is physiologically lowered to reduce fetal rejection and to allow fetal antigen tolerance. This adaptation during pregnancy may be responsible for a greater vulnerability to infections in both mother and fetus, possibly resulting in severe congenital syndrome. Most of the infections acquired during pregnancy evolve mildly in healthy adults, whereas they are a significant cause of mortality in fetuses and newborns and an important contributor to early and later childhood morbidity [1,2]. Pathogen agents include cytomegalovirus (CMV), varicella-zoster virus (VZV), rubeovirus, parvovirus B19, herpes simplex virus (HSV), Toxoplasma gondii, Treponema pallidum, Zika virus (ZIKV) and other minor aetiologies, such as adenovirus, influenza, enterovirus and lymphocytic choriomeningitis virus (LCMV) [1–12]. Ultrasound may detect different anomalies potentially related to a fetal infection, including cerebral anomalies, orbital defects, micrognathia, cardiac defects, hepatosplenomegaly, liver calcifications, other abdominal anomalies, skin and limb anomalies, edema, placental and amniotic fluid anomalies, altered Doppler analysis and intrauterine growth restriction (IUGR) [1,3–17]. Nevertheless, a normal ultrasound (US) does not exclude adverse fetal outcomes and there are limitations to the prediction of fetal involvement through precise markers. Moreover, performance depends on sonographer ability, the severity of disease, the diagnosis of fetal infection and ultrasound screening policy, which vary between countries 18. Not only the site but also the timing of the insult is critical to determine fetal abnormalities.

## 2. Materials and Methods

The selection process was conducted in duplicate and independently by two reviewers using the databases MEDLINE (accessed through PubMed), up to march 2022 We used the following keywords: "ultrasound fetal infections", "ultrasound findings", "rubella", "toxoplasma", "parvovirus", "cytomegalovirus", "herpes simplex virus", "varicella virus", "zika virus", "ultrasound fetal anomalies", "congenital infections", "ultrasound markers",

"sonographic anomalies". The studies were searched without filters or restriction on date of publication. The research was confined to papers written in English. Only studies that clearly described ultrasound features of congenital infections met the inclusion criteria for the review, so we excluded clinical studies where ultrasound features were missing or unclear. Once the reviewers had collected data independently from each report, the results were synthetized into tables. We then created one table listing ultrasound fetal anomalies with the possible corresponding pathogens involved (Table 1) and another table (Table 2) showing the associations between the pathogens and the corresponding fetal US anomalies. Ethical approval was deemed not necessary for the present narrative review.

**Table 1.** Ultrasound fetal anomalies and possible association with infectious agent.

| Anatomical Region | Ultrasound Findings | Infectious Agent |
|---|---|---|
| **CENTRAL NEVOUS SYSTEM** | VENTRICULOMEGALY | CMV |
| | | TOXO |
| | | PARVOVIRUS B19 |
| | | ZIKA |
| | | LMCV |
| | HYDROCEPHALUS | CMV |
| | | TOXO |
| | | VZV |
| | HOLOPROSENCEPHALY | CMV |
| | ANOMALIES OF CORPUS CALLOSUM | CMV |
| | | ZIKA |
| | DANDY–WALKER COMPLEX | CMV |
| | | ZIKA |
| | | RUBELLA |
| | | TOXO |
| | MICROCEPHALY | CMV |
| | | RUBELLA |
| | | ZIKA |
| | | VZV |
| | CEREBRAL CYSTS OR PSEUDOCYSTS | CMV |
| | | ZIKA |
| | PORENCEPHALY | CMV |
| | CEREBRAL CALCIFICATIONS | CMV |
| | | TOXO |
| | | VZV |
| | | ZIKA |
| | CEREBELLAR ANOMALIES | CMV |
| | | VZV |
| | | ZIKA |
| | ABNORMAL CORTICAL DEVELOPMENT | CMV |
| | PERIVENTRICULAR HYPERECHOGENICITY | CMV |
| | INTRAVENTRICULAR SINECHIA | CMV |
| | THALAMIC HYPERECHOGENICITY | CMV |
| | AGENESIS THALAMI | ZIKA |
| | BRAIN ATROPHY | ZIKA |
| | | VZV |
| | STRIATAL ARTERY VASCULOPATHY | CMV |
| | NEURAL TUBE DEFECTS | ADENOVIRUS |
| | HYDRANENCEPHALY | CMV |

**Table 1.** *Cont.*

| Anatomical Region | Ultrasound Findings | Infectious Agent |
| --- | --- | --- |
| **HEART** | CARDIAC SEPTAL DEFECTS | RUBELLA |
| | PULMONARY ARTERY STENOSIS | RUBELLA |
| | EBSTEIN'S ANOMALY | RUBELLA |
| | HEART FAILURE | PARVOVIRUS B19 |
| | | CMV |
| | CARDIOMEGALY | PARVOVIRUS B19 |
| | | RUBELLA |
| **LIVER AND SPLEEN** | HEPATOSPLENOMEGALY | RUBELLA |
| | | SYPHILIS |
| | | CMV |
| | | TOXO |
| | LIVER CALCIFICATIONS | CMV |
| | | PARVOVIRUS B19 |
| | | TOXO |
| | | VZV |
| **FACE** | ANOPHTALMY OR MICROPHTALMY | RUBELLA |
| | | SYPHILIS |
| | | VZV |
| | | INFLUENZA |
| | | PARVOVIRUS B19 |
| | CATARACTS | RUBELLA |
| | | VZV |
| | | ZIKA |
| | INTRAOCULAR CALCIFICATIONS | ZIKA |
| | MICROGNATHIA | RUBELLA |
| **SKIN, ARMS, LEGS** | SKIN LESIONS | VZV |
| | HAND/FOOT ANOMALIES | ADENOVIRUS |
| | CLUBFOOT | ZIKA |
| | RUDIMENTARY DIGITS | VZV |
| | LIMB DEFORMITIES | VZV |
| | INCREASED NT | PARVOVIRUS B19 |
| **EDEMA** | ASCITES | SYPHILIS |
| | | PARVOVIRUS B19 |
| | | CMV |
| | | TOXO |
| | HYDROTHORAX AND PERICARDIAL EFFUSION | PARVOVIRUS B19 |
| | | TOXO |
| | | CMV |
| | | SYPHILIS |
| | HYDROPS AND SKIN EDEMA | PARVOVIRUS B19 |
| | | TOXO |
| | | CMV |
| | | VZV |
| | | ADENOVIRUS |
| **PLACENTA AND AMNIOTIC FLUID** | PLACENTOMEGALY AND HYDROPIC PLACENTA | CMV |
| | | SYPHILIS |
| | | PARVOVIRUS B19 |
| | | TOXO |
| | POLYHYDRAMNIOS | SYPHILIS |
| | | VZV |
| | | CMV |
| | OLIGO-ANHYDRAMNIOS | ZIKA |
| | | CMV |
| | | VZV |

**Table 1.** *Cont.*

| Anatomical Region | Ultrasound Findings | Infectious Agent |
|---|---|---|
| **OTHER ABDOMINAL ANOMALIES** | ABDOMINAL CALCIFICATIONS | PARVOVIRUS B19 |
| | | CMV |
| | | VZV |
| | HYPERECHOGENIC BOWEL | CMV |
| | | TOXO |
| | | VZV |
| | KIDNEY'S ANOMALIES | CMV |
| | | VZV |
| **DOPPLER ALTERATIONS** | MIDDLE CEREBRAL ARTERY | CMV |
| | | RUBELLA |
| | | PARVOVIRUS B19 |
| | DUCTUS VENOSUS | CMV |
| | | PARVOVIRUS B19 |
| | UMBELICAL ARTERY | RUBELLA |
| | UTERINE ARTERIES | RUBELLA |
| **IUGR** | | ZIKA |
| | | CMV |
| | | VZV |
| | | RUBELLA |
| | | TOXO |
| | | PARVOVIRUS B19 |
| | | ADENOVIRUS |

**Table 2.** Association between the pathogens and the corresponding fetal US anomalies.

| Infectious Agent | Ultrasound Findings |
|---|---|
| **CMV** | VENTRICULOMEGALY |
| | HYDROCEPHALUS |
| | ANOMALIES OF CORPUS CALLOSUM |
| | DANDY–WALKER COMPLEX |
| | MICROCEPHALY |
| | CEREBRAL CYSTS OR PSEUDOCYSTS |
| | PORENCEPHALY |
| | CEREBRAL CALCIFICATIONS |
| | CEREBELLAR ANOMALIES |
| | ABNORMAL CORTICAL DEVELOPMENT |
| | PERIVENTRICULAR HYPERECHOGENICITY |
| | INTRAVENTRICULAR SINECHIA |
| | THALAMIC HYPERECHOGENICITY |
| | STRIATAL ARTERY VASCULOPATHY |
| | HYDRANENCEPHALY |
| | CARDIOMEGALY |
| | HEPATOSPLENOMEGALY |
| | LIVER CALCIFICATIONS |
| | ASCITES |
| | HYDROTHORAX AND PERICARDIAL EFFUSION |
| | HYDROPS AND SKIN EDEMA |
| | PLACENTOMEGALY AND HYDROPIC PLACENTA |
| | POLYHYDRAMNIOS |
| | OLIGO-ANHYDRAMNIOS |
| | ABDOMINAL CALCIFICATIONS |
| | HYPERECHOGENIC BOWEL |
| | KIDNEY ANOMALIES |
| | MIDDLE CEREBRAL ARTERY ALTERATION |
| | DUCTUS VENOSUS |
| | IUGR |

**Table 2.** *Cont.*

| Infectious Agent | Ultrasound Findings |
| --- | --- |
| **TOXO** | VENTRICULOMEGALY<br>HYDROCEPHALUS<br>MICROCEPHALY<br>CEREBRAL CALCIFICATIONS<br>HEPATOSPLENOMEGALY<br>LIVER CALCIFICATIONS<br>ASCITES<br>HYDROTHORAX AND PERICARDIAL EFFUSION<br>HYDROPS AND SKIN EDEMA<br>PLACENTOMEGALY AND HYDROPIC PLACENTA<br>HYPERECHOGENIC BOWEL<br>IUGR |
| **PARVOVIRUS B19** | VENTRICULOMEGALY<br>HEART FAILURE<br>CARDIOMEGALY<br>LIVER CALCIFICATIONS<br>ANOPHTALMY OR MICROPHTALMY<br>INCREASED NT<br>ASCITES<br>HYDROTHORAX AND PERICARDIAL EFFUSION<br>HYDROPS AND SKIN EDEMA<br>PLACENTOMEGALY AND HYDROPIC PLACENTA<br>ABDOMINAL CALCIFICATIONS<br>MIDDLE CEREBRAL ARTERY ALTERATION<br>DUCTUS VENOSUS ALTERATION<br>IUGR |
| **RUBELLA** | DANDY–WALKER COMPLEX<br>MICROCEPHALY<br>CARDIAC SEPTAL DEFECTS<br>PULMONARY ARTERY STENOSIS<br>HEART FAILURE<br>CARDIOMEGALY<br>HEPATOSPLENOMEGALY<br>ANOPHTALMY OR MICROPHTALMY<br>CATARACTS<br>MICROGNATHIA<br>MIDDLE CEREBRAL ARTERY ALTERATION<br>UMBELICAL ARTERY ALTERATION<br>UTERINE ARTERIES ALTERATION<br>IUGR |
| **VZV** | HYDROCEPHALUS<br>MICROCEPHALY<br>CEREBRAL CALCIFICATIONS<br>CEREBELLAR ANOMALIES<br>BRAIN ATROPHY<br>ANOPHTALMY OR MICROPHTALMY<br>CATARACTS<br>SKIN LESIONS<br>LIMB DEFORMITIES<br>RUDIMENTARY DIGITS<br>HYDROPS AND SKIN EDEMA<br>POLYHYDRAMNIOS<br>OLIGO-ANHYDRAMNIOS<br>ABDOMINAL CALCIFICATIONS<br>HYPERECHOGENIC BOWEL<br>KIDNEY ANOMALIES<br>IUGR |

**Table 2.** *Cont.*

| Infectious Agent | Ultrasound Findings |
|---|---|
| ZIKA | VENTRICULOMEGALY |
| | ANOMALIES OF CORPUS CALLOSUM |
| | DANDY–WALKER COMPLEX |
| | MICROCEPHALY |
| | CEREBRAL CYSTS OR PSEUDOCYSTS |
| | CEREBRAL CALCIFICATIONS |
| | CEREBELLAR ANOMALIES |
| | AGENESIS THALAMI |
| | BRAIN ATROPHY |
| | CATARACTS |
| | INTRAOCULAR CALCIFICATIONS |
| | OLIGO-ANHYDRAMNIOS |
| | IUGR |
| | HEPATOSPLENOMEGALY |
| | ANOPHTALMY OR MICROPHTALMY |
| | ASCITES |
| SYPHILIS | HYDROTHORAX AND PERICARDIAL EFFUSION |
| | HYDROPS AND SKIN EDEMA |
| | PLACENTOMEGALY AND HYDROPIC PLACENTA |
| | POLYHYDRAMNIOS |
| LCMV | VENTRICULOMEGALY |
| INFLUENZA | ANOPHTALMY OR MICROPHTALMY |
| | NEURAL TUBE DEFECTS |
| ADENOVIRUS | HAND/FOOT ANOMALIES |
| | HYDROPS AND SKIN EDEMA |

## 3. Results

*3.1. Cerebral Findings*

3.1.1. Ventriculomegaly and Hydrocephalus

It seems that infectious diseases are responsible for at least 5% of all ventriculomegaly diagnosed prenatally 18. Ventriculomegaly is common in cases of fetal CMV [1,3–10,18–22], Toxoplasma gondii [1,5,6,23], parvovirus B19 [19], Zika virus [11,13,15,24–30], LCMV [31,32] and VZV infection [33].

Gestational age at the time of infection is the most important factor influencing the risk of vertical transmission of CMV infection and the possible evolution of brain anomalies. Guerra et al. reviewed 600 cases of maternal primary CMV infection during pregnancy. The most common anomalies were hyperechogenic bowel and ventriculomegaly, both in infected and uninfected fetuses. Seven infected fetuses were reported as having isolated ventriculomegaly and their outcomes were: no symptoms, mild hepatitis, mild ventriculomegaly at birth and monolateral hearing loss al 1 year, severe ventriculomegaly with lissencephaly and calcifications at birth, psychomotor delay and hearing loss by first year. The authors reported ultrasound sensitivity of only 15%, which increased when the infection was diagnosed by PCR in the amniotic fluid [18]. Recently, Leruez-Ville analyzed 82 pregnancies complicated by CMV fetal infection, diagnosed by analysing either amniotic fluid or fetal blood. In the case of US severe anomalies of central nervous system, the authors reported a negative predictive value of symptoms at birth or termination of pregnancy of 93%. Therefore, 7% of fetuses were missed for severe symptoms in the third trimester [7]. Nevertheless, some CMV-related anomalies, including chorioretinitis, petechiae and neurodevelopment defects, are not evident by US. Other typical abnormalities, such as ventriculomegaly, echogenic bowel and IUGR, can be detected by US [8]. Lazzarotto and colleagues hypothesized hypoxic damage when they found cytomegalic cells and cortex laminar necrosis together with 80% of placental villi necrotic or hydropic. These findings suggest a viral direct effect and/or a placenta-mediated insufficiency at the base of brain anomalies [3]. Dogan reviewed eight pregnancies complicated by CMV infection diagnosed

by amniocentesis and reported intracranial anomalies and ventriculomegaly in 8/8 and 6/8 fetuses, respectively [4]. The presence of cerebral anomalies is still the main prognostic factor, despite several articles investigating the time between CMV infection and US signs of fetal infection [34]. The most frequent anomalies detected by US were summarized by Benoist: ventriculomegaly, periventricular hyperechogenicity or halo, microcephaly, calcifications, periventricular pseudocysts, intraventricular synechiae, abnormalities of the cortical development, the cerebellum and the posterior fossa, cerebral haemorrhage, IUGR, splenomegaly, hepatomegaly, hyperechogenic bowel, cardiomegaly, isolated pleural or pericardial effusion and ascites or hydrops [34]. Usually, CMV infection involves diffusely fetal organs due to its tropism for endothelial cells [20]. Frequently, ventriculomegaly is part of a pattern of atrophy and loss of cerebral volume and, even if it is not specific to CMV fetal infection, though it is the second most common finding, it is indicative of it when found with concomitant microcysts or periventricular echogenic foci [35].

US can detect congenital toxoplasmosis, including ventriculomegaly and/or hydrocephalus and other intra- and extracranial anomalies; nevertheless, these are only suggestive of fetal infection [1,5,6,23]. The visualization of hydrocephalus is not only diagnostic of fetal infection but also suggestive of grave fetal involvement. The reported sensitivity of US in the diagnosis of congenital toxoplasmosis is 45%, while specificity is 99,8% [5].

Parvovirus may be responsible for fetal ventriculomegaly, as described in a retrospective study including 141 fetuses with isolated mild ventriculomegaly (imVM). Pasquini found 6/129 and 6/135 fetuses to be infected by parvovirus B19 and CMV, respectively. Prenatal and postnatal tests confirmed infection only in two cases: one by parvovirus B19 and one by CMV. In the first case, imVM regressed before birth, whereas in the second one it was stable during pregnancy [19].

A widespread epidemic of ZIKV infection was reported in 2015 in South and Central America and the Caribbean. An apparent increase in microcephaly in fetuses born to mothers infected with ZIKV has been noticed [11,13,24,31]. Mlakar et al. described a case-report of a woman living in Brazil who had a fever with rash at the end of the first trimester of pregnancy. Microcephaly with cerebral and placental calcifications was found by US at 29 weeks of gestation. Moderate ventriculomegaly was also reported at 32 weeks of gestation. The woman opted for TOP and fetal autopsy confirmed antenatal anomalies (microcephaly, agyria, hydrocephalus and brain calcifications) and ZIKV was found in the fetal brain by PCR analysis [11]. Oliveira reported two cases of fetal microcephaly possibly related to ZIKV. One case presented severe unilateral ventriculomegaly. Some features resembled those of CMV and Toxoplasma gondii fetal infection, though with a more severe pattern and a lack of nodules, respectively [24]. Brasil published a cohort study in which 42 pregnant women infected by ZIKV underwent US. They reported 12/42 abnormal US results with only one case with non-isolated ventriculomegaly and the pregnancy still viable at the time of publication 13. Recently, a review found sufficient evidence to conclude that Zika virus is a cause of congenital abnormalities [26]. The designation "congenital Zika syndrome" (CZS) has been chosen to identify a new congenital syndrome, including microcephaly, the primary neurological complication, and different neurological anomalies, i.e., ventriculomegaly, hydrocephalus, holoprosencephaly, microcephaly, lissencephaly, polymicrogyria, agyria and brain calcifications [27,30]. CZS includes also musculoskeletal (arthrogryposis, scoliosis and hip dislocation), ocular, craniofacial, genitourinary, pulmonary and other anomalies [28–30].

Two articles published in 2007 and 2014 described some alterations present in children with congenital LCMV infection, including ventriculomegaly and hydrocephalus [31,32]. LCMV is an enveloped single-stranded RNA virus belonging to the Arenaviridae family. Even if it is referred to neuroimaging performed during the postnatal period, it is important to consider this association, even if there are less than 100 cases reported in the literature. This virus is thought to be selectively neurotrophic in case of antenatal infection, affecting the brain and retina in 87.5% of cases. Typically, prenatal infection involves brain parenchyma without systemic effects; on the contrary, postnatal infection affects meninges

and the choroid plexus [32]. LCMV infection may lead to microcephaly, neuronal migration anomalies, periventricular calcifications, pachygyria, porencephalic cysts, periventricular cysts, hydrocephalus, seizures and subsequent neurodevelopment disability.

Hydrocephalus may present as part of congenital varicella syndrome [33]. When these alterations are found, the sonographer must evaluate the possible presence of other defects. In case of apparently isolated ventriculomegaly, there may be underlying cerebral malformations, such as lissencephaly or destructive lesion (periventricular leukomalacia). Hydrocephalus has been reported also in cases of CZS [27,29].

### 3.1.2. Cysts and Pseudocysts

Cerebral cysts and pseudocysts are substantially related to CMV infection [4,7,9,10,22, 35–37]. CMV typically affects the germinal zone and pseudocysts correspond to necrosis of this region and the ependymal border [10]. They can be initially seen as a vacuolization and be variously located, but most frequently they are located in the region adjacent to the anterior temporal lobes. In the same area, white matter abnormalities are often present, together being specific for fetal CMV infection [35,36]. Other areas which might be involved are within the frontoparietal white matter and adjacent to the occipital poles of the lateral ventricles [35,37]. Malinger et al. reviewed the sonograms of eight fetuses diagnosed with CMV infection and found five cases of echogenic intraparenchymal foci and ventriculomegaly. Again, echogenic intraparenchymal foci and ventriculomegaly were present in five fetuses. Other signs were abnormal sulcation (4 cases) and periventricular cysts (3 cases) [22].

Leruez-Ville distinguished between periventricular cystic lesions (severe CNS anomaly) and subependymal and choroid plexus cysts (non-severe CNS anomalies); thus, clinically normal neonates with pseudocysts in the germinal matrix were considered asymptomatic. There are few studies associating porencephaly with CMV infection [7].

### 3.1.3. Periventricular Hyperechogenicity and Ventricular Synechia

Malinger observed 4/8 fetuses with proved CMV infection and periventricular altered echogenicity: two fetuses had echogenic and well-defined anomalies with cystic formations, whereas in the other two the periventricular tissue protruded through an undefined ependyma. Three fetuses also presented intraventricular adhesions, seen as thin strands of tissue that crossed the ventricles [22]. Intraventricular synechia are a non-specific finding of CMV infection and may be reported also in case of intraventricular haemorrhage or ventriculitis [35].

Similarly, Dogan reported eight cases of diagnosed congenital CMV infection. Periventricular hyperechogenicity was the leading sign (7/8 cases). Four fetuses also had intraventricular synechia—a striking feature of the infection [4].

### 3.1.4. Microcephaly

Microcephaly is usually determined at birth, but it may be diagnosed prenatally with ultrasound at 18–20 weeks. Among the infectious agents, CMV is frequently involved when microcephaly is found at US scan. Other responsible agents include VZV, rubeovirus, Zika virus and Toxoplasma gondii [1,6,11,13,23–30,38].

The linkage between CMV infection during pregnancy and microcephaly is supported by many sources in the literature. Fink et al., in 2010, described CMV-linked brain atrophy, such as a combination of microcephaly, ventriculomegaly and/or generalized loss of cerebrum or cerebellum volume, especially if the infection occurred early in pregnancy. Microcephaly was found in 27% of patients, regardless of the time of infection, and other ultrasound findings include a wide opening of the cerebellar folia and an expansion of the posterior fossa. The postnatal diagnosis may be confirmed with magnetic resonance imaging (MRI) or computerized tomography (CT) [35].

Rubella congenital syndrome can include microcephaly. One review defines congenital rubella syndrome (CRS), which includes microcephaly, microphthalmia, pulmonary artery stenosis, cardiac septal defects and hepatosplenomegaly [38].

ZIKV has a strong neurotrophism causing microcephaly and cataracts [11,13,24–30]. The study by Brasil et al. included 88 enrolled pregnant women, among whom 72 were positive for ZIKA virus. Twelve fetuses had abnormal ultrasound findings: five IUGR, four cerebral calcifications, four cases of abnormal flow in the umbilical or cerebral arteries, two oligohydramnios/anhydramnios, two cases with cysts and four calcifications [13]. Meaney-Delman et al. reviewed information on ZIKV, including its effects during pregnancy and strategies of prevention, to guide obstetrical cares for pregnant women residing in areas affected by ZIKV transmission and who were travelling or had previously travelled to those areas [25]. They reported microcephaly, brain atrophy, ventriculomegaly and intracranial calcifications in neonates positive for ZIKV infection, yet the full spectrum of fetal anomalies is unknown. Furthermore, an increased number of fetal and neonatal cerebral anomalies (microcephaly, destruction of cerebral structures, cerebellar agenesis and ventriculomegaly) were noticed in 2014–2015 in French Polynesia. In addition, the number of newborns with severe microcephaly reported in Brazil appears to be much greater than expected, considering hospital-based data for previous years. The case-report by Mlakar et al. reported microcephaly with cerebral and placental calcifications found by US at 29 weeks of gestation [11]. Until the last few years, no fetal cases of ZIKV infection had been reported, probably because of underreporting of cases, the early acquisition of immunity in endemic areas or the initial rarity of the disease. It must be considered that cases of microcephaly currently reported represent only the more severely affected children and that infants with less severe disease have not yet been diagnosed [24,27].

### 3.1.5. Calcifications

Intracranial calcifications are very frequent in fetal CMV infections, occuring in 34–70% of cases, appearing as echogenic foci with or without acoustic shadowing [30]. In case of CMV infection they are usually distributed periventricularly [1,4–10,23,35,39], whereas Toxoplasma gondii tends to cause randomly distributed intracranial calcifications [1,5,6,23].

In case of CMV infection, calcifications are commonly thick and periventricular, but they can be also faint and punctate, involving the brain parenchyma and basal ganglia. Their absence does not exclude CMV fetal infection. The presence of cerebral calcifications is more strongly associated with neurodevelopment delays than other US anomalies and is anyhow suggestive of severe fetal involvement [5,35].

Congenital varicella syndrome is characterized by certain typical features (limb hypoplasia, rudimentary digits, microcephaly and microphthalmia) and intracranial calcifications [23].

Cerebral calcifications have been found in ZIKV fetal infection [11,13,25,27,29,30], in particular in the cortex and subcortical white matter in the frontal, parietal and occipital lobes [11,27].

### 3.1.6. Anomalies of the Middle Lane

Anomalies of the middle lane include holoprosencephaly and abnormalities of the corpus callosum (CC). CMV [7,10,22,24,40] and Zika virus [13,25,27,29,30] are the main pathogens involved. Holoprosencephaly is a spectrum of cerebral abnormalities resulting from incomplete cleavage of the forebrain. According to the degree of forebrain cleavage, three types can be distinguished: the alobar type, the most severe one, is characterized by the fusion of the thalami and a monoventricular cavity; the semi-lobar type includes a partial posterior separation of the ventricles and cerebral hemispheres with incomplete fusion of the thalami; the lobar type presents a normal separation of the ventricles and the thalami in the absence of the septum pellucidum. Holoprosencephaly may have different causes, including CMV as infectious etiology [40]. The anomalies of the CC can also result from the fetal transmission of the virus.

Malinger reviewed US findings for eight fetuses with proved CMV infection and found two cases with non-isolated hypoplastic CC and other anomalies, including ventriculomegaly, sulcation and gyral abnormal patterns, signs of striatal artery vasculopathy, echogenic intraparenchymal foci, intraventricular adhesions, periventricular pseudocysts and cerebellar and cisterna magna abnormalities [22]. Alby analyzed 138 cases of fetuses with CC malformation (complete agenesis, dysgenesis, hypoplasia and dysplasia) and found one case of CMV infection [24]. Picone described US features in congenital CMV infection and found one case of CC agenesis, one case of CC hypoplasia and one case of cerebellar hypoplasia [10].

### 3.1.7. Anomalies of the Posterior Fossa

Dandy–Walker complex includes the Dandy–Walker malformation (enlarged posterior fossa with complete or partial agenesis of the cerebellar vermis), Dandy–Walker variant (normal posterior fossa with partial agenesis of the cerebellar vermis) and mega cisterna magna, defined as a vertical distance >10 mm between the vermis and the inner border of the skull (with normal cerebellar vermis and fourth ventricle). CMV [4,7,22], ZIKV [36] and rubeovirus [12] infections can be involved in these anomalies.

CMV can lead to various degrees of cerebellar dysplasia. Dogan reported eight cases of diagnosed congenital CMV infection: he found three cases of mega cisterna magna (14–17 mm), in addition to cerebral abnormalities; two cases of vermian defects; one case of cerebellar calcifications; and one case of cerebellar cysts [4].

ZIKV has been associated with vermian dysgenesis and enlarged cisterna magna [41].

A recent case report of congenital rubella describes the agenesis of the inferior cerebellar vermis and an anomalous pulmonary venous drainage. It was hypothesized that the agenesis of the inferior cerebellar vermis was related to cellular damage caused by inflammation and incomplete inferior fusion [12].

Cerebellar anomalies also include cerebellar agenesis, atrophy, hypoplasia and dysplasia. The main infectious cause is CMV [3,9,10,22], with minor contributions of VZV [14] and ZIKV [25,27,30].

Moreover, US findings of intracranial haemorrhage and hippocampal dysplasia have been linked to congenital CMV infection [35].

### 3.1.8. Abnormal Cortical Development

CMV has been often related to abnormal cortical development [3,4,7,9,10,22,35,36,42–46]. Migrational anomalies have been reported in 10% of fetuses with CMV infection. The most common anomalies are lissencephaly, pachygyria and polymicrogyria [2,35,36]. Lissencephaly is characterized by a smooth brain surface with the absence of sulcation. The cortex is often thickened, but if the infection is associated with neuronal loss the cortex can be thin. Pachygyria is characterized by broad gyri with partial sulcation. This finding is suggestive of early fetal infection and the outcome is worse than polymicrogyria [43]. Concomitant hypoplastic CC was found in two fetuses with abnormal sulcation and CMV infection [22]. There are few studies reporting polymicrogyria diagnosed by US [22,44], which consists of focal or diffuse multiple small abnormal gyri, thickened and nodular cortex and irregular grey–white matter junctions. Dhombres reported a rare case of polymicrogyria CMV-related diagnosed by US at 27 weeks of gestation. The fetus did not show cortical hyperechogenicity, but an over-folded cortex was clearly visible due to the enlargement of pericerebral space secondary to microencephaly. Other US findings of infectious fetopathy were absent; nonetheless, the coexistence of polymicrogyria, microencephaly, bilateral opercular dysplasia, septate ventricles and periventricular cysts was particularly evocative of an infectious disease [42]. Diagnosis is easier during the second trimester than during the third, when secondary sulci have developed, and dedicated TA and TV neurosonography could be considered equal to MRI in the diagnosis of fetal brain anomalies [45].

Recently, some authors have described these anomalies in cases of ZIKV infection contributing to the designation of CZS [27,29]. Due to its deleterious manifestations, it has been proposed to include them in the list related to those of TORCH agents [29].

Schinzencephaly is a rare cortical malformation that manifests as a grey matter-lined cleft extending from the ependyma to the pia mater, which may be part of CMV congenital infection features [35]. It can be open-lip (the cleft is widely patent) or closed-lip (the walls of the cleft are closely apposed).

Cortical dysplasia together with white matter disease was reported in one fetus with congenital CMV infection [46].

### 3.1.9. Other Anomalies

Other anomalies are thalamic hyperechogenicity, agenesis thalami, brain atrophy, neural tube defects, widened subarachnoid space, hydranencephaly, vascular anomalies and focal cerebral destruction. CMV is the possible infectious cause of almost all these anomalies, as reported in various articles already mentioned [4,7,9,10,22], and in one case a report described a widened subarachnoid space with documented CMV infection [47].

Brain atrophy has been related to ZIKV [25,30] and VZV infection [6,23]. ZIKV has also been found as a new cause of agenesis of the thalami and a possible cause of destructive lesions [25,27].

Neural tube defects (NTDs) include spina bifida, anencephaly and encephalocele, and adenovirus has been found as a possible cause [15]. Hydranencephaly is associated with fetal CMV infection [7]. This virus has been associated with signs of striatal artery vasculopathy [22]. Lenticulostriate vasculopathy is visualized as unilateral or bilateral curvilinear echogenic streaks within the basal ganglia and thalami, possibly denoting a mineralizing vasculopathy of the lenticulostriate vessels. It is reported in 27% of patients with CMV infection [37], though this is a non-specific finding, as it was reported also in the case of prenatal drug exposure, twin–twin transfusion syndrome, cromosomopathy, congenital heart disease, toxoplasmosis and human immunodeficiency virus (HIV) infection [35,48].

Reported in the literature is a single case of West Nile virus fetal infection showing chorioretinitis and focal cerebral destruction, which raised concern for teratogenicity; subsequent studies have not suggested an increased risk [25,41,49]. Furthermore, VZV reactivation may cause intrauterine encephalitis, leading to destructive and inflammatory lesions [14].

### 3.2. Face
### 3.2.1. Orbital Defects

Defects of the eyes, such as anophthalmia and microphthalmia, can result from a congenital infection, particularly in cases of rubeovirus [5,6,23,38]. Embryologically, the eyes originate from three germ layers: the neuroectoderm (optic vesicle), the neural crest cells (anterior chamber) and the ectoderm (lens placode). The neuroectoderm and mesoderm contribute to the closure of the optic fissure. This multiple embryological derivation causes the great variability in the phenotypes and aetiologies of congenital ocular defects.

### 3.2.2. Anophthalmia and Microphthalmia

Anophtalmia means the complete atrophy of the eyes, optic nerves and/or chiasma, while microphtalmia refers to a smaller size of the eyeballs. Sonography is able to identify prenatally three orbital structures: lens, pupils and optic nerves.

Rubeovirus is most commonly responsible for these anomalies, though other agents include Treponema pallidum, VZV, influenza, parvovirus B19 and ZIKV.

CRS includes microphthalmia as a common anomaly together with cataracts [5,6,23,38].

Treponema Pallidum usually causes other types of ocular lesion, such as chorioretinitis, interstitial keratitis, iridocyclitis and pigment epithelial dystrophies, but we found a case report that describes fetal bilateral microphtalmia caused by transmission of gestational syphilis [50].

VZV might cause microphthalmia. The virus and its vesicles usually locate following one or several nerve distributions [14,23]. Disseminated VZV infection can interfere with the normal development of the fetal nervous system. Indeed, the virus is neurotropic and can affect the central, peripheral or autonomic systems. In case of involvement of the optic tract, the result could be microphtalmia or chorioretinitis, while the deeper damage to the fetal brain can cause microcephaly.

Influenza and parvovirus B19 are less well-established teratogens. Busby et al.,found that there was a temporal link between the prevalence of anophthalmia and microphthalmia in England and the epidemics of these infections. They also describe a positive association between parvovirus B19 counts and the risk of severe anophthalmos and microphthalmos [51]. As for influenza virus, the study speculates that the fever caused by the inflammatory response to the virus could be central in the genesis of fetal ocular anomalies. On the other hand, Coxsackie virus seems not to be involved in optical defects. Previous studies described an association between Coxsackie B virus, congenital cardiac inflammation and spontaneous abortion. No association was described between Coxsackie A virus and adverse pregnancy outcomes. Indeed, Coxsackie viruses are supposed to be stronger in eliminating selectively affected fetuses in early pregnancy [51].

Microphthalmia was reported in several cases of congenital Zika virus infection [24,25,27,30]. In a recent analysis of 35 babies born from women who lived in or visited Zika virus endemic sites during their pregnancy, almost one out of five newborns had ocular defects: 3% had microphthalmia and 8% had abnormal funduscopic examination [52].

### 3.2.3. Cataracts and Optical Nerve Anomalies

Possible infectious causes are rubeovirus [5,6,23,24,38], VZV [14,23] and Zika virus [24,25,27,30].

CRS was first described by an ophthalmologist, but only one case of prenatal eye disease has been reported. This syndrome commonly includes also IUGR and cardiac and neurologic abnormalities. The virus may stay in the aqueous humour until adulthood, causing cataracts, microphthalmia, glaucoma and chorioretinitis; however, the leading cause of congenital cataracts is idiopathic [5]. Direct cytopathic effects of the virus are probably responsible for cataract development [24]. Yazigi et al. reviewed 32 cases of CRS between 1991 and 2014 diagnosed during the first trimester between 1991 and 2014. Seventeen of these 32 fetuses showed 56 different anomalies, including two ocular defects: cataracts and microphthalmia [38]. We must consider that most ocular anomalies are missed during the antenatal US examination, while they are frequently found in postnatal studies.

Fetal varicella syndrome (FVS) may affect the central and autonomic nervous systems, causing eye, skin or limb defects. More precisely, in the literature, VZV has been found to be associated with: microphthalmia, chorioretinitis, cataracts, opaque cornea, optic nerve atrophy or hypoplasia and Horner syndrome [14].

Moreover, cataracts and microphthalmia are described in cases of congenital ZKV infection [24,25,27].

### 3.2.4. Intraocular Calcifications

The presence of hyperechoic areas in the eye indicates calcifications, which have been recently found in fetuses of mothers with ZKV infection [24,25,30]. Few reports link the flaviviruses to fetal brain and eyes insults [11].

### 3.2.5. Craniofacial Anomalies

Micrognathia is usually related to genetic syndromes, chromosomal disorders and teratogenical drugs. The literature lacks descriptions of craniofacial defects of infective origin, except for one case report describing an atypical presentation of rubeovirus infection with several fetal deformities that included micrognathia [12]. Rubeovirus inhibits cell growth, produces cytolysis and interferes with blood supply. The widespread virus causes

cytopathic damage to blood vessels that leads to vasculitis, endothelial necrosis, and cardiac lining. This mechanism can explain the cardiovascular and nervous anomalies brought on by the persistence of the rubeovirus [12].

Craniosynostosis results from the premature fusion of the cranial sutures and it has been described in relation to ZIKV fetal infection [27]. Additionally, it represents a cause of microcephaly.

### 3.3. Heart

#### 3.3.1. Cardiac Septal Defects

The most commonly described cardiac septal defects due to congenital infection are those associated with CRS [5,38]. We found a literature review concerned with congenital anomalies due to CRS [38] that reported three cases of ventricular septal defects and two cases of atrial septal defects out of 32 fetuses with CRS diagnosed prenatally, along with 69 (23%) postnatal descriptions of septal defects out of 290 newborns with CRS. One case-report on congenital rubella describes an atrial septal defect with right-to-left flow, together with dilatation of the right atrium and ventricle, tricuspid regurgitation, pulmonary hypertension and patent ductus arteriosus with bidirectional flow [12,38].

#### 3.3.2. Pulmonary Artery Stenosis

A few cases of pulmonary artery stenosis associated with CRS have been recognized in utero, mostly severe [5,6,38]. Yazigi's review reports 81 cases (28%) of pulmonary artery stenosis that may be accessible to prenatal diagnosis out of 290 newborns with CRS [38].

#### 3.3.3. Ebstein's Anomaly

We found a case report about this rare ultrasound finding associated with CRS [53].

#### 3.3.4. Heart Failure

Several mechanisms related to vertical infections can lead to intrauterine heart failure, including severe anaemia and direct cardiac infections. Pregnancies complicated by parvovirus B19 infection can manifest with hemodynamic alterations detectable with ultrasonography and Doppler studies [16].

Carraca et al. describe early signs of heart failure linked to parvovirus infection in early pregnancy: an increase in nuchal translucency (NT) and reversed flow in the ductus venosus (DV) during atrial contraction. Carraca noticed that NT calculation and DV Doppler analysis are useful indicators of fetal anaemia and cardiac dysfunction: in cases of fetal anaemia, blood viscosity decreases, causing an increased fetal cardiac output that leads to a bigger venous return and cardiac preload. A recent study explains how in late pregnancy an increased cardiac output leads first to hypervolemia and hydrops and then, if fetal compensatory mechanisms are extinguished, to cardiac failure. In the first trimester, however, cardiac failure can appear earlier during the infection process because cardiac afterload is appreciably greater than in later gestation [54].

#### 3.3.5. Cardiomegaly

Congenital myocarditis can manifest with cardiomegaly, which is often associated with pericardial effusion and ascites. Early CMV infections can cause cardiac changes, including cardiomegaly [8,39]. The ultrasound images of the hypertrophic heart walls infected by CMV may include calcifications, evident as hyperechoic areas in the context of the cardiac muscle [16]. With similar mechanisms, myocardial infection by parvovirus B19 can manifest with cardiomegaly, in addition to the hemodynamic alterations mentioned above [1,17]. One case of cardiomegaly associated with CRS has been described in the literature [55].

*3.4. Liver and Spleen*

3.4.1. Hepatosplenomegaly

There are multiple causes of this condition, such as immune and non-immune hydrops, metabolic disorders, fetal infection, haemangioma or hepatoblastoma. The infectious pathogens involved are CMV [1,3,4,6–10,23], rubeovirus [5,12,38], Treponema pallidum [6,23,56,57] and Toxoplasma gondii [1].

CMV congenital infection is symptomatic in 11% of cases and hepatosplenomegaly is a frequent finding (jaundice, petechiae, hepatosplenomegaly, microcephaly and intracranial calcification are observed in 75% of symptomatic infants), whereas hepatomegaly is one of the prenatal detectable US anomalies [8].

A recently published review focuses specifically on the signs of congenital rubella syndrome accessible to prenatal diagnosis. The author classified the postnatal abnormalities according to their accessibility to antenatal diagnosis: hepatosplenomegaly is reported to be present in 2/32 fetuses with CRS diagnosed prenatally and 55/1109 children with CRS [38].

A prospective study analyzed 24 women with untreated syphilis during pregnancy and reported a 66% rate of fetuses with hepatomegaly [56]. A more recent study evaluated sonographic features of congenital syphilis and described how they changed after antibiotic treatment: 73 (30%) had evidence of fetal syphilis on initial ultrasound scan and hepatomegaly was present in 79% of fetuses. Infant outcomes were available for 173 deliveries: of these, 32 infants (18%) were diagnosed with congenital syphilis. Congenital syphilis was more common when antenatal ultrasound abnormalities were present (39% vs. 12%) [57].

3.4.2. Liver Calcifications

Calcifications can be visualized during routine ultrasound as echogenic foci in the parenchyma or capsules in the liver. CMV [1,7,10,58], Toxoplasma gondii [6,23], parvovirus B19 [59] and VZV [1,5] can be involved. An isolated liver calcification does not have great significance, particularly if infective diseases or chromosomal disorders are excluded. A prospective study published in 2002 including 61 pregnant women with fetal liver calcifications analyzed the aetiologies of these anomalies: 40 out of 61 patients had additional fetal abnormalities; 21 out of 61 cases of intrahepatic calcifications were isolated; 11 out of 61 patients (18%) had abnormal karyotypes. Two patients had intrauterine infections: one patient had cytomegalovirus and one patient had parvovirus B19 infection [58].

*3.5. Abdominal Anomalies*

3.5.1. Abdominal Calcifications and Hyperechogenic Bowel

The presence of hyperechogenic bowel or abdominal hyperechoic foci are non-specific ultrasonographic findings which might occur during fetal infection with parvovirus B19 [17], CMV [1,3,6,8–10,23,39] and VZV [1,14].

Puccetti and colleagues analyzed the outcome of 63 pregnancies complicated by maternal parvovirus B19 infection. Hyperechogenic bowel was present in 3/63 infected fetuses: in two cases it resolved spontaneously, whereas the other showed intra-abdominal calcifications at the 30th week and a meconium peritonitis at birth which needed surgical treatment [17].

Moreover, the possible involvement of Toxoplasma gondii was noted [6].

3.5.2. Kidney Anomalies

Hyperechogenic kidneys and hydronephrosis have been found to be possible signs of CMV [1,10] and VZV [6] fetal infection, respectively.

*3.6. Skin, Arms and Legs*

3.6.1. Skin Lesions, Rudimentary Digits and Clubfoot

The pathogens which may be implicated are VZV [14,23,33], adenovirus [15] and ZIKV [13,27]. VZV usually produces skin lesions following dermatome pattern distribu-

tions, supporting the hypothesis of a herpes zoster in utero. These are usually retracted scars, while sometimes they remain active [14].

Adams et al. elaborated a retrospective observational study on women who underwent amniotic fluid karyotyping and viral PCR testing for anamnestic reasons or ultrasound anomalies. Adenovirus and cytomegalovirus were the most frequently isolated anomalies [15]. Adenovirus is the primary associated pathogen; nevertheless, it can also be found in normal pregnancies and therefore the moment of the vertical transmission of the disease is crucial. The potential mechanisms consist of an inhibition of embryonic neuronal migration due to direct effects or indirect effects due to maternal fever.

In addition, ZKV could be a possible aetiology of club foot, also called congenital talipesequinovarus [13,27].

### 3.6.2. Limb Anomalies

Ultrasounds can detect hypoplasia or atrophy of limbs. The most documented infectious cause is VZV [1,6,14,23,24,33]. The virus can affect the nervous system diversely: the involvement of the spinal cord and somatic ganglia can cause denervation and hypoplasia of the limbs [24]. As previously mentioned, fetal lesions are segmental [14], including musculoskeletal (hypoplasia of muscles and/or bones and malformed digits) and nervous system anomalies. Auriti's case report described a woman who contracted chickenpox in the 12th week of gestation [33]. At 13 weeks of gestation, elevated serum IgG and IgM levels were found by enzyme-linked immunosorbent assay (ELISA). Congenital varicella syndrome (CVS) was diagnosed prenatally by ultrasound evaluations performed at the 29th and 34th weeks of gestation showing limb deformities: the right femur was considerably shorter than the left femur (5.89, −2 SD for 34 weeks gestational age, vs. 6.97 cm).

CZS includes different musculoskeletal anomalies, such as arthrogryposis, fetal akinesia deformation sequence and related malformations, bilateral acetabular dysplasia and amyoplasia of the lower limbs [27,29,30]. Probably these alterations are secondary to ZIKV-induced brain injury rather than a direct cytopathic effect [27].

### 3.7. Edema

### 3.7.1. Nuchal Translucency (NT)

NT thickness is increased with chromosomal abnormalities, cardiac defects and many genetic syndromes. Parvovirus B19 is the main factor responsible for increased NT at ultrasound [17,54]. Carraca et al. described a fetus with increased NT and reversed a-wave in the ductus venosus (DV) caused by parvovirus B19 infection at 11 weeks, with fetal demise at 13 weeks [54]. The mechanism by which parvovirus infects erythroid precursors is mediated by a cellular receptor expressed by erythroblasts, megakaryocytes, erythrocytes, synoviums, placental tissue, fetal myocardium and endothelial cells. Its tropism inhibits haematopoiesis and its effects show up in the first and second trimester. Increased NT is associated with myocardial dysfunction and fetal anaemia as well as alterations of the DV [54].

### 3.7.2. Ascites

The pathogens involved in this condition are: CMV, parvovirus B19, Treponema pallidum and Toxoplasma gondii [1,6–10,17,23,39,56,57,59]. Ascites and cardiomegaly are manifestations of heart failure following fetal parvovirus B19 infection [59].

Ascites is one of the most common ultrasound features of fetal syphilis, together with other sings of hydrops. After specific treatment, it appeared to resolve, along with polyhydramnios and MCA Doppler anomalies. The persistence of ascites and hepatomegaly has been associated with treatment failure [57].

### 3.7.3. Hydrothorax and Pericardial Effusion

Toxoplasma gondii and CMV can cause both anomalies. Hydrothorax can be present also during parvovirus B19 infection [4,7,9,10,17,23,59]. Puccetti et al. reported one case of

isolated hydrothorax in a fetus diagnosed with parvovirus B19 infection, in which the fluid was drained after spontaneous birth without consequences for the newborn [17].

### 3.7.4. Hydrops and Skin Edema

Skin edema is mainly associated with congenital syphilis and CMV fetal infection [1,5–7,9,23,60]. Hydrops aetiologies also include parvovirus B19 (first reported in 1984 [59]), VZV, Toxoplasma gondii and ZIKV [1,5–7,13–15,17,23,27,59,60]. Other possible consequences are severe fetal anaemia, cardiac failure and myocarditis [59]. Fetal anaemia leads to an increased cardiac preload with consequent hypervolemia, which is the primary cause of hydrops [54]. After its diagnosis, fetal anemia can be treated with intrauterine blood transfusion [17]. Enders et al. established that there is a significant B19-associated risk of fetal hydrops and fetal death which seems to be limited to vertical infections that happen between the 9th and the 20th weeks [61].

### 3.8. Placental and Amniotic Fluid Anomalies

### 3.8.1. Placentomegaly and Placental Calcifications

Placentomegaly reflects a placental thickness or placental extent expressing fetal disease. The increase in placental size may be homogenous or heterogeneous depending on the cause. The maximum thickness considered normal at any stage in pregnancy is often considered 4 cm.

CMV is often implicated in placental anomalies, such as placentomegaly and placental calcifications. CMV first infects the placenta and then the fetus; therefore, placental thickening might precede fetal infection. La Torre analyzed 92 cases of primary CMV infection (47 cases with fetal transmission and 43 cases without fetal involvement) in a comparison with 73 seropositive women without gestational infection [62]. Thirty-two women received CMV hyperimmune globulin (HIG) and this was associated with a statistically significant reduction in placental thickness, which might have been due to the expression of suppressed viral infection and reduced inflammation. It is possible that CMV-related placental insufficiency could be responsible for fetal and neonatal disease [62]. Increased placental weight was previously reported to be associated with CMV congenital infection [63]. Garcia described specific placental anomalies, such as diffuse vascular inflammation, villitis, necrosis, syncytial knots and calcifications, resembling IUGR cases. Hypoxia could cause fibrinoid deposition and small vascularized villi leading to placentomegaly. Other pathogens causing placentomegaly are Treponema pallidum [56,57], parvovirus B19 [1] and Toxoplasma gondii [6,23]. Placentomegaly is a common initial finding in cases of fetal syphilis together with hepatomegaly, followed by hematologic dysfunction, ascites and fetal immunoglobulin M (IgM) production [56]. The commonest ultrasound finding is placentomegaly, found in 27% of 73 women with diagnosed syphilis and abnormal ultrasound scans out of 235 women with silent syphilis [57]. All women received benzathine penicillin G. Serial US was performed after initial abnormal US until resolution or delivery, and the data revealed that, after treatment, the findings that are thought to upsurge early, including placentomegaly, persisted the longest. Hepatomegaly was the last anomaly to resolve [57].

Parvovirus B19 fetal infection was found to be associated with placental hydrops in 7/63 cases, all of which were characterized also by fetal anaemia, IUGR and hydrops [17]. Six cases ended with TOP or intrauterine death, while only case resolved after intrauterine transfusion with delivery at term [17].

### 3.8.2. Polyhydramnios and Oligo-/Anhydramnios

Both conditions are associated with VZV [5,14] and CMV [6,7,9,10,39] infections. Fetal syphilis includes various anomalies: polyhydramnios, hepatomegaly, placentomegaly, ascites and abnormal middle cerebral artery Doppler assessment [57]. A recent preliminary report describes a few cases of Zika infection during pregnancy which may evolve in a mild way in the mother and cause severe fetal involvement on the other hand. Microcephaly and IUGR were the most common US findings; in addition, one fetus presented

oligo-/anydramnios [13]. Current articles report both polyhydramnios and oligohydramnios during ZIKV fetal infection [64].

### *3.9. Doppler Alterations*

### 3.9.1. Middle Cerebral Artery (MCA)

CMV infection may cause fetal hypoxia and shows an increased diastolic flow in the MCA [16]. Reductions in the resistance index (RI) and pulsatility index (PI) of the fetal MCA are also described in association with congenital rubella syndrome, especially if the infection occurs before 20 weeks of gestation, causing severe damage to the villous vessels and consequent vascular flow centralization in the fetus [16]. In cases of parvovirus B19 fetal infection, peak systolic velocity in the MCA can be in the normal range but there can be a reversed diastolic flow secondary to increased intracranial pressure. One such case, without any evidence of fetal anaemia, was described by Zajicek et al. in 2010 [64]. In cases of fetal anaemia, the peak systolic velocity will be higher than normal, as described in Puccetti's cohort of 63 pregnant women with B19 infection: all anaemic fetuses had MCA peak systolic velocity values more than 1.8 times the median [17]. Meanwhile, Rac analyzed a cohort of syphilis-infected fetuses: of the fifty-two women who had MCA peak systolic velocity (PSV) measured, 17 (33%) were found to have elevated velocities [57].

### 3.9.2. Ductus Venosus (DV)

When CMV infection induces hepatic dysfunction, Doppler analysis may show hemodynamic alterations, such as reflux in the ductus venosus during atrial contractions [16]. Reverse flow in the DV is also found in fetal anaemia, when fetal cardiac output is increased because of lower blood viscosity, leading to a higher venous return and cardiac preload. Parvovirus B19 was found to be responsible for a reverse a-wave in the DV associated with an increased NT in a first trimester-infected fetus described by Carraca et al. in 2011 [54].

### 3.9.3. Umbilical Artery (UA)

An alteration of the RI and PI of the umbilical artery can be detected by Doppler velocimetry in case of congenital infection, indicating a high peripheral resistance. Congenital rubella infection can cause damage to the villous vessels, leading in the worst cases to placental haemorrhage and necrosis; the result is an augmentation of the resistive and pulsatility indices in the umbilical artery [16].

### 3.9.4. Uterine Artery

In case of early transplacental infection, the resultant villitis affects the trophoblastic invasion of the spiral arteries, causing higher RI and PI of the uterine arteries, which mirrors an increased peripheral resistance. This can be seen in case of CRS, especially if the infection is transmitted early in pregnancy [16].

## 4. Discussion

The ultrasound diagnosis of fetal infections is of great interest and importance for pregnancy management and therapy—to inform parents, to guide the diagnosis after birth and to organize the follow-up. Many articles have been published on this topic concerning specific fetal infections. The present review evaluates the specific organ involvements which may correlate with antenatal infectious diseases to guide differential diagnosis and prenatal care.

Indeed, many infectious diseases during pregnancy have been found to be related to specific US findings. Many studies in the literature focus on cerebral findings, such as ventriculomegaly or hydrocephalus, cerebral cysts and pseudocysts, periventricular hyperechogenicity and ventricular synechia, microcephaly, cerebral calcifications, anomalies of the middle lane and of the posterior fossa or abnormal cortical development [19–30]. Furthermore, some studies describe anomalies in developments such as orbital defects,

anophthalmia or microphthalmia, cataracts and optical nerve anomalies, intraocular calcifications or craniofacial anomalies, such as micrognathia or craniosynostosis [5,6,23,27,38].

Heart defects, such as cardiac septal defects, pulmonary artery stenosis and Ebstein's anomaly, are mainly linked to CRS [39,65]. Heart failure is mainly described in association with parvovirus B19 infection, while cardiomegaly can be caused by CMV, parvovirus B19 or CRS [53].

A few studies have described abdominal defects diagnosed at US that include hepatospenomegaly, liver calcifications, abdominal calcifications and hyperechoic bowel, or kidney anomalies, such as hyperechoic kidneys or hydronephrosis [1,3,4,6–10,12,17,23,38].

Rarely, some congenital infections can cause skin lesions, rudimentary digits and clubfoot and other limb anomalies [1,5–7,9,14,23,24,33,60].

Edema, such as increased NT thickness, ascites, hydrothorax or pericardial effusion, hydrops or skin edema can also result from specific congenital infections [17,54].

Defects of placental development, such as placenomegaly or placental calcifications, or increased and decreased amniotic fluid amount can be associated with specific infections [6,23,56,57].

In the end, maternal and fetal Doppler alterations can reflect early transplacental infection and they may be signs of severe fetal hypoxic changes due to congenital infections [16,17,54,55].

Our study was conceived to help clinicians and sonographers to direct the diagnosis of fetal infection for confirmation by maternal serologies and by a careful analysis of possible maternal exposures to infectious agents. Once a diagnosis is made, the transmission of the infection to the fetus can be confirmed by amniocentesis, if feasible and accepted by the mother. This review did not aim to correlate ultrasound characteristics with fetal prognoses, as that is a task that will require further specific studies.

**Author Contributions:** P.G.: conceived and supported the study; D.M.: critically revised the final version of the manuscript; F.R.: carried out the investigation and wrote the manuscript; E.S.: carried out the investigation and wrote the manuscript; G.S.: critically revised the final version of the manuscript; R.V.: conceived the study and supported the investigation. All authors have read and agreed to the published version of the manuscript.

**Funding:** This research received no external funding.

**Institutional Review Board Statement:** Ethical review and approval were waived for this study, since the article does not contain any studies with human participants or animals performed by any of the authors.

**Informed Consent Statement:** Patient consent was waived since this study is based upon data collection available in the published literature.

**Acknowledgments:** This work was performed at the Arcispedale Sant'Anna, University of Ferrara. All authors have participated in the work and take public responsibility for appropriate portions of the content.

**Conflicts of Interest:** The authors certify that they have no affiliations with or involvement in any organization or entity with any financial interest or non-financial interest in the subject matter or materials discussed in this manuscript.

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
