# Peer review of "Ultrasound Findings of Fetal Infections: Current Knowledge"

_2673-3897, doi:10.3390/reprodmed3030016_

Round 1
Reviewer 1 Report
The present review reports about ultrasound findings of fetal infections.
Major points/revisions: There are no major points to be corrected
Abstract: The abstract is well written and shows briefly the aim and results of this review.
Introduction section: The main idea that not only the site but also the timepoint of detection of an abnormality in fetal ultrasound is mentioned. This implements the consideration of embryological development.
Material – Methods: This chapter is relatively weak. There are no standards named that have to be used for a broad analysis of current scientific literature, like Prisma 2020. The idea to correlate ultrasound findings and infectious agent in table1 is very good. The table itself should be reshaped to get a clearer overview.
Results: This chapter is well written and brings together what authors selected out of the literature.
Discussion: The discussion is very short and doesn`t reflect the arguments out of the informations from the results part. Here some conclusions would be worthful, for example - how to handle the ultrasound findings in the consultation situation? or - Is there an algorithm foreseeable to conclude out of a combination of findings to a pathway of diagnostics and prognostic criteria?
Author Response
Reply to Reviewer 1:
Comments and Suggestions for Authors
The present review reports about ultrasound findings of fetal infections.
- Major points/revisions: There are no major points to be corrected
- Abstract: The abstract is well written and shows briefly the aim and results of this review.
- Introduction section: The main idea that not only the site but also the timepoint of detection of an abnormality in fetal ultrasound is mentioned. This implements the consideration of embryological development.
- Material – Methods: This chapter is relatively weak. There are no standards named that have to be used for a broad analysis of current scientific literature, like Prisma 2020. The idea to correlate ultrasound findings and infectious agent in table1 is very good. The table itself should be reshaped to get a clearer overview.
Thank you for your suggestions, we added information in MATERIALS & METHODS about the selection process and synthesis methods, according to PRISMA 2020 checklist. However, this is a descriptive review that lacks statistical analysis, sub-group analysis or meta-regression.
- Results: This chapter is well written and brings together what authors selected out of the literature.
- Discussion: The discussion is very short and doesn`t reflect the arguments out of the informations from the results part. Here some conclusions would be worthful, for example - how to handle the ultrasound findings in the consultation situation? or - Is there an algorithm foreseeable to conclude out of a combination of findings to a pathway of diagnostics and prognostic criteria?
Thank you for your suggestions, we lengthen the discussion trying to synthetize information out of the results. The prognosis of each infectious disease is not argued in this review, that focuses on diagnosis and US findings.
Reviewer 2 Report
The subject addressed by the authors is of great interest to gynaecologists and useful in daily practice. However, I would have a few mentions: I suggest moving the tables from the methods to the results section. The information is quite hard to follow and some divided tables by authors' results would be useful. There are large paragraphs without bibliographic indexes, and the references are not sufficient for an exhaustive review. I recommend adding more bibliographic sources. The discussion part at the end does not have its place. I recommend replacing it with an improved conclusion section that supports the topic more strongly.
Author Response
Reply to reviewer 2:
- Comments and Suggestions for Authors: The subject addressed by the authors is of great interest to gynaecologists and useful in daily practice. However, I would have a few mentions:
- I suggest moving the tables from the methods to the results section. Ok
- The information is quite hard to follow and some divided tables by authors' results would be useful. There are large paragraphs without bibliographic indexes, and the references are not sufficient for an exhaustive review. I recommend adding more bibliographic sources. The discussion part at the end does not have its place I recommend replacing it with an improved conclusion section that supports the topic more strongly.
Thank you for your suggestions, we lengthen the discussion trying to synthetize information out of the results